# Impaired Vestibulo-Ocular Reflex on Video Head Impulse Test in Superior Canal Dehiscence: "Spontaneous Plugging" or Endolymphatic Flow Dissipation?

Andrea Castellucci [1,*], Pasquale Malara [2], Salvatore Martellucci [3], Mohamad Alfarghal [4], Cristina Brandolini [5], Gianluca Piras [6], Enrico Armato [7], Rosanna Rita Ruberto [8], Pasquale Brizzi [8], Livio Presutti [5] and Angelo Ghidini [1]

1 ENT Unit, Department of Surgery, Azienda USL—IRCCS di Reggio Emilia, 42123 Reggio Emilia, Italy; angelo.ghidini@ausl.re.it
2 Audiology & Vestibology Service, Centromedico, 6500 Bellinzona, Switzerland; pasmalara@gmail.com
3 ENT Unit, Santa Maria Goretti Hospital, Azienda USL di Latina, 04100 Latina, Italy; dott.martellucci@gmail.com
4 Otorhinolaryngology—Head and Neck Section, Surgery Department, King Abdulaziz Medical City, Jeddah 21556, Saudi Arabia; audio1972@gmail.com
5 Otorhinolaryngology and Audiology Unit, IRCCS Azienda Ospedaliero-Universitaria di Bologna, Policlinico S. Orsola-Malpighi, 40138 Bologna, Italy; cristina.brandolini@aosp.bo.it (C.B.); livio.presutti@unibo.it (L.P.)
6 Department of Otology and Skull Base Surgery, Gruppo Otologico, Casa Di Cura Privata "Piacenza" S.P.A., 29121 Piacenza, Italy; gianlu.piras@libero.it
7 Faculty of Medicine, University of Lorraine, 54000 Vandoeuvre-lès-Nancy, France; armato.otovest@gmail.com
8 Audiology and Ear Surgery Unit, Azienda USL—IRCCS di Reggio Emilia, 42123 Reggio Emilia, Italy; rosannarita.ruberto@ausl.re.it (R.R.R.); pasquale.brizzi@ausl.re.it (P.B.)
* Correspondence: andrea.castellucci@ausl.re.it; Tel.: +39-0522-296273; Fax: +39-0522-295839

**Abstract:** Surgical plugging of the superior semicircular canal (SSC) represents an effective procedure to treat disabling symptoms in superior canal dehiscence (SCD), despite resulting in an impaired vestibulo-ocular reflex (VOR) gain for the SSC. On the other hand, SSC hypofunction on video head impulse test (vHIT) represents a common finding in patients with SCD exhibiting sound/pressure-induced vertigo, a low-frequency air–bone gap (ABG), and enhanced vestibular-evoked myogenic potentials (VEMPs). "Spontaneous canal plugging" has been assumed as the underlying process. Nevertheless, missing/mitigated symptoms and/or near-normal instrumental findings would be expected. An endolymphatic flow dissipation has been recently proposed as an alternative pathomechanism for SSC VOR gain reduction in SCD. We aimed to shed light on this debate by comparing instrumental findings from 46 ears of 44 patients with SCD exhibiting SSC hypofunction with postoperative data from 10 ears of 10 patients with SCD who underwent surgical plugging. While no difference in SSC VOR gain values was found between the two groups ($p = 0.199$), operated ears developed a posterior canal hypofunction ($p = 0.002$). Moreover, both ABG values ($p = 0.012$) and cervical/ocular VEMP amplitudes ($p < 0.001$) were significantly higher and VEMP thresholds were significantly lower ($p < 0.001$) in ears with SCD compared to operated ears. According to our data, canal VOR gain reduction in SCD should be considered as an additional sign of a third window mechanism, likely due to an endolymphatic flow dissipation.

**Keywords:** superior canal dehiscence; vestibulo-ocular reflex; third window; plugging; video head impulse test; vestibular-evoked myogenic potentials; conductive hearing loss; air–bone gap

## 1. Introduction

The introduction of superior canal dehiscence (SCD) syndrome among inner ear disorders and the description of its related clinical and instrumental features has allowed physiologists to demonstrate how a bony hole in the vestibular partition of the otic capsule

can account for a low-impedance pathway for sounds and pressure stimuli, resulting in a range of peculiar audio-vestibular symptoms and signs consistent with a third mobile window mechanism (TMWM). This condition has been widely accepted as the pathomechanism accounting for autophony, own-body sound hyperacusis, pulsatile tinnitus, conductive hearing loss (CHL) with a low-frequency air–bone gap (ABG), sound- and/or pressure-induced nystagmus, and an enhanced vestibular-evoked myogenic potential (VEMP). Similarly, the onset of eye movements aligning with the plane of the dehiscent superior semicircular canal (SSC) in response to loud sounds (Tullio phenomenon), sudden pressure changes (Hennebert sign), and skull vibrations has been ascribed to the same pathomechanism [1–8]. Namely, an SCD acts as a third mobile window and allows the dissipation of acoustic energy from the cochlear to the vestibular partition, thus leading to an impaired cochlear sensitivity for air-conducted (AC) sounds and a reduced threshold for bone-conducted (BC) stimuli [4–6]. The amount of endolymphatic flow that is shunted away from the cochlea is capable of activating SSC afferents leading to the nystagmus aligning with the SSC plane in response to sound and pressure stimuli [1,2,6,7]. On the other hand, the bony defect at the SSC reduces the labyrinthine impedance and increases the sensitivity of the otolith organs to AC and BC stimuli, generating abnormal VEMPs with enhanced amplitudes and lowered thresholds [3,6,7]. The improvement of the aforementioned symptoms and the restoration of instrumental abnormalities after surgical plugging of the canal, irrespective of the technique adopted, strongly support this assumption. In fact, despite developing a vestibulo-ocular reflex (VOR) gain impairment for the affected canal, SSC plugging has been demonstrated to be an effective treatment for disabling symptoms due to a TMWM through a restoration of a physiological two-window system, thus resulting in ABG closure, normalization of electrophysiological measurements, and the receding of sound/pressure-induced vertigo [1,3,6,9–25]. On the contrary, a different mechanism other than TMWM has been implied to explain the reduced VOR gain values for the affected SSC after head impulses, as is often measured in patients with wide-sized SCD either with a magnetic-scleral search coil or, more commonly, through the video head impulse test (vHIT) [2,9,26–29]. In fact, the growing accessibility of canal VOR gain measurements in the high-frequency domain promoted by the recent introduction of the vHIT in clinical practice has basically replaced the hardly accessible search coil technique [30], allowing the detection of peculiar lesion patterns, including selective canal loss [26,27,31–37]. In particular, a spontaneous plugging process through a progressive herniation of the middle fossa dura into the SSC through the dehiscence has been proposed as the underlying pathomechanism for a selective blockage of endolymphatic flows within the affected SSC, thus behaving as a natural equivalent of surgical canal plugging [2,9,27]. Even though a spontaneous dural prolapse within the SSC has been anecdotally confirmed through imaging [38,39], if that were true in all SCD cases exhibiting an impaired function for the affected canal, patients should either lack symptoms or only develop mitigated symptoms, besides exhibiting near-normal instrumental measurements, as expected in patients following a surgical plug. Conversely, according to the correlation analysis available in the literature comparing SCD size and symptoms/signs, subjects with wide-sized SCD seem to behave oppositely. In fact, despite the results of the correlation analysis possible in part mismatching, which was likely due to the heterogenous cohorts, variable mean SCD sizes, and different modalities for the measure of the dehiscence width, most studies agree that the SCD size positively correlates with particularly intense audio-vestibular symptoms [40–46]. Additionally, a direct positive correlation has been registered between SCD size and ABG and both cervical (cVEMP) and ocular VEMP (oVEMP) amplitudes, whereas the width of the dehiscence seems to inversely correlate with the cVEMP/oVEMP threshold and SSC VOR gain values [2,9,10,26,29,42,44,46–51]. An endolymphatic flow dissipation during head impulses has been recently proposed as an alternative pathomechanism accounting for an "apparent" SSC VOR gain impairment in patients with SCD presenting with "active" symptoms consistent with a TMWM [52]. The aim of this study is to shed light on the actual mechanism accounting for a VOR gain impairment in SCD. Therefore, we compared the instrumental

data of the ears with SCD exhibiting SSC hypofunction on vHIT to post-operative data of ears with SCD that were submitted to surgical plugging.

## 2. Materials and Methods

This study was approved by our Institutional Review Board (approval number 143/2019/Oss/AOUBo) and was conducted according to the tenets of the Declaration of Helsinki. We performed a retrospective review of clinical-instrumental data of all patients who were referred at our institutions from January 2003 to December 2018 and were diagnosed with unilateral or bilateral SCD according to the Barany Society diagnostic criteria [53]. Then, we reviewed the clinical-instrumental data of all patients who received a surgical SSC plugging for disabling symptoms at our institutions in the same period. All patients underwent the same detailed assessment including pure tone audiometry, AC and/or BC cVEMPs, AC and/or BC oVEMPs, and VOR gain assessment for all semicircular canals with the vHIT. Patients who were operated on received an audio-vestibular assessment prior to surgery, within 2 weeks after surgery, and at 6 months. Only subjects with complete clinical and instrumental data were recruited. For correlation analyses between instrumental findings of not-operated SCD and SCD that was operated on, we selected two distinct groups of ears:

1. Ears with SCD exhibiting a VOR gain impairment for the affected SSC were included in the "not-operated SCD" group;
2. Ears with SCD that were operated on through a surgical plugging due to disabling symptoms were included in the "post-operative SCD" group. In this case, the instrumental measurements used for the correlation analysis refer to data collected at 6-months after surgery.

Ears with other concurrent inner/middle ear abnormalities (mainly with Meniere's disease (MD), vestibular schwannoma, otosclerosis, or middle ear infections) were not admitted to the study because of possible interference in data collection. We also excluded from the "post-operative SCD" group the patients who did not develop improvement of symptoms following SCD surgery, or who exhibited recurrence of symptoms after surgery, to avoid including patients who might have received incomplete SSC plugging.

### 2.1. Pure Tone Audiometry and Impedance Audiometry

Pure tone audiometry was performed over the frequency range of 125 to 8000 Hz for AC and 250 to 4000 Hz for BC in a soundproof room using standard clinical procedures. Appropriate masking was used for BC testing and, when needed, for AC. Pure tone average for AC was calculated across 125–8000 Hz, whereas mean BC threshold was calculated across 250–4000 Hz. The average low-frequency ABG was derived by subtracting the BC threshold from the AC threshold for each individual frequency including 250, 500, and 1000 Hz. Tympanometry and ipsi/contralateral acoustic reflexes were tested. In case of type-A tympanogram, peak compliance value was calculated.

### 2.2. VEMPs

Potentials were recorded using either an Epic Plus evoked potentials system (Labat, Mestre, Italy) with a two-channels averaging capacity or a 2-channel evoked potential acquisition system (Viking, Nicolet EDX, CareFusion, Heidelberg, Germany). All unrectified electromyographic (EMG) potentials were amplified and the signal was bandpass-filtered from 10 Hz to 1 kHz. Potentials were recorded with active surface electrodes placed symmetrically over the most prominent part of each sternocleidomastoid muscle (SCM), reference electrodes placed on each clavicle, and a ground electrode that was applied over the upper sternum. For AC cVEMP testing, the patient was supine with the head turned on the non-stimulated ear side to get a tonic contraction of SCM ipsilateral to the stimulus presentation. As for BC cVEMPs, with stimulus at the midline forehead (Fz), tonic SCM activation on both sides was obtained by instructing the patient to raise their head constantly by approximately 30 cm from the supine position for at least 15 s. oVEMPs

were recorded using a differential bipolar montage, with surface electrodes placed 1 cm (positive) and 3 cm (negative) beneath each lower eyelid. A ground electrode was placed over the sternum. Patients were required to direct their gaze upwards for the duration of the recording. Ongoing EMG activity of tested muscles was visually monitored on an oscilloscope to ensure sufficient muscle contraction, so that the acquisition was conducted only in case the biofeedback-based "greenlight" signal (i.e., "adequate contraction") was active. As for both AC cVEMPs and oVEMPs, tone bursts (TB) were delivered unilaterally through a pair of insert earphones. Stimulus parameters were as follows: frequency, 500 Hz; duration, 8 ms (raise/fall 2 ms, plateau 4 ms); starting intensity, 120 dB SPL; and stimulation rate, 5 Hz. Threshold was obtained by decreasing in steps of 10 dB or increasing in steps of 5 dB from 120 dB SPL depending upon the presence or absence of potentials, respectively (maximal released intensity of 125 dB SPL). The lowest stimulus intensity at which a clear and repeatable biphasic wave could be detected was considered as threshold. As for BC cVEMPs and oVEMPs, 500 Hz TB (8 ms, 1.0 V, 0.6 A) stimuli were delivered to the midline (Fz) using a hand-held minishaker with an attached perspex rod (type 4810, Bruel&Kjaer P/L, Naerum, Denmark). Vibratory stimulation was varied in intensity and amplification through a power amplifier (type 2718, Bruel&Kjaer P/L, Denmark). BC oVEMPs were also tested even after impulsive Fz stimuli using 0.5 ms clicks with positive polarity (intensity 0.7 V, 0.6 A). At least 70 sweeps were averaged for each trial and responses were reproduced at least twice. As for cVEMPs, the first biphasic response (p13–n23) on the ipsilateral SCM to the stimulated side was analyzed (ipsilateral response) by calculating the peak-to-peak amplitude between p13 and n23 waves. For oVEMPs, the first negative (n1) and the first positive (p1) waves were analyzed by calculating the peak-to-peak amplitude. Responses recorded under the patient's right eye were interpreted as the activation of his left utriculus and vice versa (crossed response). In case of no reliable response (absent VEMPs), for the purpose of statistical analysis, a threshold value corresponding to 130 dB SPL (i.e., 5 dB higher than the maximal intensity delivered by the instrument) and an amplitude of 0.5 $\mu$V for cVEMPs and 0.2 $\mu$V for oVEMPs (i.e., the least amplitude values that can be detected in each acquisition window) were assigned.

### 2.3. vHIT

All subjects were assessed with the vHIT to measure the VOR gain for each semicircular canal. VOR gain for horizontal (HSC), superior (SSC), and posterior semicircular canals (PSC) in response to high-frequency head stimuli was tested using an ICS video-oculographic system (GN Otometrics, Taastrup, Denmark). Passive, unpredictable 5°–20°, 50°–250°/s, and 750°–5000°/s$^2$ head impulses were delivered manually on the plane of the horizontal and vertical canals (RALP and LARP planes) while the patient was asked to keep looking at an earth-fixed target, as reported by Halmagyi et al. [30]. At least 15 stimuli were delivered for stimulating each canal and averaged to obtain the corresponding mean VOR gain. VOR gain values <0.8 for HSC and <0.7 for ASC and PSC with corrective saccades (overt and/or covert) were considered pathological. Data corresponding to VOR gain values for SSC, HSC, and PSC for each pathologic ear were considered in the statistical analysis.

### 2.4. Imaging

All patients in the study group underwent temporal bones high-resolution CT (HRCT) scans. Images were acquired parallel to the orbito–meatal axial plane and reconstructed in the coronal plane. Parasagittal reconstructions of the SSC along the Pöschl plane for each temporal bone were also obtained. Dehiscence length (size) was evaluated by measuring the distance between bony remnants of the canal opening on reconstructions along the Pöschl plane.

### 2.5. Surgical Plugging

All patients who were operated on due to disabling symptoms received SSC plugging via a transmastoid approach in the ear with SCD. In the cases with bilateral SCD, we performed the surgical procedure only in the most symptomatic ear. Once a mastoidectomy was performed with identification of the sigmoid sinus, middle fossa tegmen, and HSC, the middle fossa dura was skeletonized. Subarcuate artery was used as surgical landmark for the identification of the SSC. Bony SSC was blue-lined with a small cutting burr, opened either through 1 or 2 fenestrations and filled with different materials (bone wax, bone dust, and strips of fascia) to achieve a complete canal occlusion as proposed in literature [3,11,16]. Particular care was taken to avoid breaching the membranous labyrinth or suctioning over the exposed portion of the canal. Following canal occlusion, the SSC was reinforced with temporalis fascia, bone pâté, and fibrin sealant. Post-operative oral steroid tapering for 7 days and 48 mg Betahistine a day for 2 weeks was administered in all cases.

### 2.6. Statistical Analysis

Quantitative variables were checked for normal distribution using both Kolmogorov–Smirnov and Shapiro–Wilk tests. Continuously distributed variables were described by mean $\pm$ 1 standard deviation (SD) or by median, interquartile range (IQR), and range. The Student t-test or the Mann–Whitney U test were used to compare median values of instrumental variables. Statistical significance was presented as *p*-value and it was assumed that a null hypothesis could be rejected at $p < 0.05$. Statistical analyses were performed using IBM SPSS ver. 20.0 (IBM Corp., Armonk, NY, USA).

### 3. Results

Forty-four patients (21 males, 23 females, median age 65, IQR 15, range age 56 years old) with SCD exhibiting an SSC VOR gain reduction on the vHIT were recruited, including 2 patients with bilateral SCD with a VOR gain impairment for both SSCs. Therefore, 46 ears (23 right, 23 left) with SCD and SSC hypofunction were included in the "not-operated SCD" group. The median SCD size was 4.1 mm (IQR 1.5, range 4.6 mm); the membranous SSC was in contact with the superior petrosal sinus (SPS) in three cases. All details regarding the instrumental findings of each ear included in this group with corresponding descriptive statistics are reported in Table 1. Ten subjects (4 males, 6 females, median age 49, IQR 26, range age 55 years old) who underwent a successful SSC plugging for disabling symptoms and met the inclusion criteria were recruited. Autophony, fullness, and pulsatile tinnitus were the most frequent disabling auditory symptoms (9/10), whereas sound and/or pressure-induced vertigo represented the most invalidating vestibular symptoms (9/10). In total, 70% of cases presented with low-frequency ABG, while all of them exhibited nystagmus in response to skull vibrations and abnormal VEMPs. Four patients presented with normal SSC VOR gain measures pre-operatively and all of them exhibited an SCD size $\leq$ 3.0 mm (see Table 2 for details concerning the pre-operative symptoms and signs). All of them were operated on only in one side, as both patients with bilateral SCD only received surgery in the most symptomatic ear. Therefore, 10 ears (3 right, 7 left) with surgically plugged SSC were selected and included in the "post-operative SCD" group. The median SCD size was 3.1 mm (IQR 2.0, range 3.2 mm) in this subgroup. In one case, the dehiscence was caused by a deep groove of the SPS. A mild transient post-operative sensorineural hearing loss (SNHL) could be registered in 3/10 patients, while 2/10 patients exhibited a severe post-operative hearing impairment that was treated with tapered oral steroids and hyperbaric oxygen therapy with only a partial recovery. All patients experienced unsteadiness in the post-operative period and exhibited transient spontaneous nystagmus either in the horizonal plane or aligning with the plane of the plugged SSC, while half of them (5/10) developed benign paroxysmal positional vertigo (BPPV) involving the ipsilateral PSC, which resolved after appropriate canalith repositioning maneuvers. Six patents developed a transient global semicircular canal hypofunction on the vHIT that eventually resulted in a selective SSC impairment for the affected SSC in all cases and

in a persistent slight PSC functional loss in six cases (see Table 3 for details regarding the post-operative symptoms and signs recorded within the first 2 weeks from surgery). Detailed information for the instrumental findings at 6 months for each ear included in this group with corresponding descriptive statistics can be found in Table 4. Patients recruited in the "not-operated SCD" group were significantly older than those who received canal plugging ($p = 0.036$), while the mean SCD size was similar among the subgroups ($p = 0.181$) (Figure 1). We correlated the mean values of the instrumental data between the ears included in the "not-operated SCD" group and those recruited in the "post-operative SCD" group at 6 months from surgery. The first group exhibited a significantly larger ABG compared to the ears that were operated on ($p = 0.012$) (Figure 2A). While there was no statistically significant difference among the two groups in terms of SSC VOR gain ($p = 0.199$) and HSC VOR gain values ($p = 0.74$), the mean VOR gain value for the PSC was significantly lower in the "post-operative SCD" group compared to the "not-operated SCD" group ($p = 0.002$) (Figure 2B–D). As for AC cVEMPs and oVEMPs (measured in 56/56 ears and 54/56 ears, respectively), the mean threshold values were significantly lower and amplitudes significantly higher in the "not-operated SCD" group ($p < 0.000$). Only 34/56 patients were tested using cVEMPs and oVEMPs to BC stimuli. The "not-operated SCD" ears exhibited significantly higher amplitudes for both BC-cVEMPs and BC-oVEMPs ($p = 0.001$ and $p < 0.001$, respectively) compared to the ears in the "post-operative SCD" group (Figures 3 and 4).

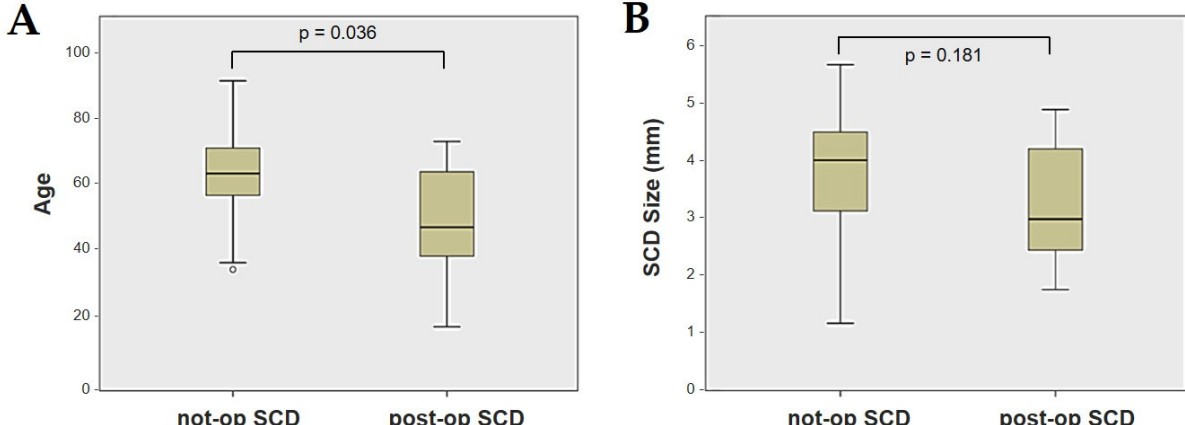

**Figure 1.** Box plots correlating patients' age (**A**) and SCD size (**B**) between not-operated ears with SCD and ears with SCD after surgical plug, with corresponding *p* value. Not-op: not-operated ears. Post-op: post-operative ears. SCD: superior canal dehiscence.

**Table 1.** Demographics and instrumental findings of each ear of the "not-operated SCD" group included in the analysis (n = 46).

| Ear | Age | Sex | SCD Side | SCD Size (mm) | ABG (dB) | VOR Gain on vHIT | | | AC cVEMPs | | BC cVEMPs | AC oVEMPs | | BC oVEMPs | |
|---|---|---|---|---|---|---|---|---|---|---|---|---|---|---|---|
| | | | | | | SSC | HSC | PSC | Thres (dB SPL) | Ampl (µV) | Ampl (µV) | Thres (dB SPL) | Ampl (µV) | TB-Ampl (µV) | Click-Ampl (µV) |
| 1 | 59 | M | L | 4.6 | 20.00 | 0.47 | 1.04 | 0.73 | 90 | 262.5 | 271.0 | 95 | 6.4 | 9.8 | 16.6 |
| 2 | 36 | M | R | 3.7 | 33.33 | 0.47 | 1.15 | 0.87 | 80 | 237.5 | 201.5 | 80 | 51.4 | 25.8 | 27.0 |
| 3 | 54 | M | L | 2.2 | 23.33 | 0.61 | 1.03 | 0.90 | 85 | 223.0 | 268.0 | 85 | 28.6 | 31.0 | 12.4 |
| 4 | 38 | M | L | 3.8 | 25.00 | 0.56 | 1.13 | 0.79 | 80 | 150.5 | 217.5 | 90 | 49.6 | 47.8 | 48.2 |
| 5 | 40 | M | R | 3.2 | 11.67 | 0.69 | 1.05 | 0.98 | 90 | 281.5 | 224.0 | 80 | 30.0 | 19.2 | 11.8 |
| 6 | 61 | F | L | 4.3 | 28.33 | 0.49 | 1.31 | 0.70 | 65 | 202.0 | 277.0 | 75 | 55.6 | 40.6 | 40.0 |
| 7 | 58 | F | R | 4.9 | 20.00 | 0.50 | 1.06 | 0.82 | 85 | 105.5 | 203.0 | 95 | 7.8 | 5.4 | 16.2 |
| 8 | 68 | F | R | 3.9 | 30.00 | 0.69 | 1.15 | 0.80 | 90 | 204.5 | 132.5 | 90 | 11.4 | 9.6 | 7.0 |
| 9 | 59 | F | R | 3.5 | 23.33 | 0.67 | 1.34 | 1.14 | 95 | 272.5 | 176.0 | 100 | 19.2 | 1.6 | 6.8 |
| 10 | 58 | M | L | 4.9 | 10.00 | 0.67 | 0.88 | 0.79 | 90 | 90.7 | - | 95 | 23.2 | - | - |
| 11 | 66 | F | R | 4.1 | 13.33 | 0.45 | 0.99 | 0.75 | 80 | 151.5 | 94.0 | 90 | 8.2 | 4.6 | 5.0 |
| 12 | 74 | M | L | 4.2 | 28.33 | 0.46 | 0.83 | 0.74 | 85 | 99.0 | 155.0 | 85 | 33.4 | 14.0 | 7.2 |
| 13 | 70 | F | R | 2.7 * | 35.00 | 0.39 | 1.01 | 0.68 | 85 | 169.0 | 295.5 | 95 | 26.8 | 13.4 | 9.2 |
| 14 | 58 | F | L | 2.2 | 6.67 | 0.67 | 1.12 | 0.76 | 110 | 53.5 | 105.0 | 130 | 0.2 | 8.8 | 9.2 |
| 15 | 70 | F | L | 4.4 | 41.67 | 0.51 | 1.15 | 0.79 | 90 | 146.5 | 211.5 | 105 | 6.2 | 7.8 | 15.6 |
| 16 | 40 | F | R | 4.3 | 23.33 | 0.64 | 1.03 | 0.71 | 70 | 186.5 | 184.5 | 75 | 13.4 | 12.8 | 19.0 |
| 17 | 48 | F | R | 1.2 | 31.67 | 0.43 | 0.98 | 0.83 | 70 | 333.0 | - | 75 | 67.8 | - | - |
| 18 | 72 | F | R | 4.9 | 20.00 | 0.52 | 0.85 | 0.73 | 80 | 66.5 | 221.0 | 95 | 16.6 | 3.2 | 5.4 |
| 19 | 77 | M | R | 3.9 | 38.33 | 0.54 | 0.80 | 0.80 | 90 | 151.5 | 93.0 | 100 | 8.4 | 11.2 | 7.4 |
| 20 | 73 | F | L | 3.2 | 28.33 | 0.28 | 0.92 | 0.70 | 90 | 69.5 | 200.0 | 95 | 28.4 | 25.0 | 16.2 |
| 21 | 74 | F | R | 1.5 | 28.33 | 0.60 | 1.15 | 0.72 | 120 | 30.0 | 172.5 | 105 | 7.6 | 13.2 | 14.0 |
| 22 | 44 | M | R | 2.9 | 20.00 | 0.66 | 1.05 | 0.88 | 80 | 295.0 | 248.5 | 75 | 31.8 | 15.8 | 23.2 |
| 23 | 76 | M | L | 2.6 | 25.00 | 0.54 | 1.17 | 1.16 | 130 | 0.5 | 67.5 | 130 | 0.2 | 2.2 | 9.0 |
| 24 | 72 | F | L | 3.8 | 26.67 | 0.54 | 0.90 | 0.79 | 85 | 197.5 | 89.0 | 80 | 57.2 | 22.4 | 18.6 |
| 25 | 59 | M | R | 4.2 | 43.33 | 0.43 | 0.85 | 0.91 | 95 | 168.0 | 81.0 | 100 | 19.2 | 20.0 | 28.2 |
| 26 | 38 | F | R | 5.1 | 25.00 | 0.51 | 1.07 | 0.85 | 70 | 185.0 | 227.5 | 70 | 26.8 | 14.6 | 23.4 |
| 27 | 39 | M | L | 4.1 | 13.33 | 0.42 | 1.01 | 0.83 | 75 | 309.0 | 275.5 | 85 | 20.6 | 8.2 | 6.4 |
| 28 | 75 | M | L | 4.7 | 30.00 | 0.39 | 0.98 | 0.71 | 85 | 143.5 | 81.5 | 85 | 38.2 | 13.4 | 9.4 |
| 29 | 74 | M | L | 3.5 | 19.67 | 0.41 | 0.84 | 0.71 | 70 | 219.0 | - | 95 | 25.5 | - | - |
| 30 | 65 | F | R | 1.5 * | 13.33 | 0.47 | 0.94 | 0.95 | 110 | 146.0 | - | 95 | 70.9 | - | - |
| 31 | 58 | M | L | 4.6 | 15.00 | 0.66 | 0.87 | 0.88 | 70 | 163.0 | - | 85 | 33.1 | - | - |
| 32 | 67 | F | L | 3.4 | 18.33 | 0.39 | 0.82 | 0.73 | 80 | 270.0 | - | 95 | 33.0 | - | - |
| 33 | 69 | F | R | 5.0 | 11.67 | 0.49 | 1.51 | 0.96 | 80 | 312.0 | - | 95 | 22.8 | - | - |
| 34 | 69 | F | L | 4.4 | 11.67 | 0.53 | 1.47 | 0.86 | 80 | 316.0 | - | 95 | 29.6 | - | - |
| 35 | 51 | M | L | 5.8 | 30.00 | 0.13 | 1.14 | 0.78 | 70 | 188.0 | - | 85 | 44.0 | - | - |
| 36 | 60 | M | L | 5.0 | 18.33 | 0.17 | 0.98 | 0.91 | 100 | 76.0 | - | 100 | 41.0 | - | - |

**Table 1.** *Cont.*

| Ear | Age | Sex | SCD Side | SCD Size (mm) | ABG (dB) | VOR Gain on vHIT | | | AC cVEMPs | | BC cVEMPs | AC oVEMPs | | BC oVEMPs | |
|---|---|---|---|---|---|---|---|---|---|---|---|---|---|---|---|
| | | | | | | SSC | HSC | PSC | Thres (dB SPL) | Ampl (µV) | Ampl (µV) | Thres (dB SPL) | Ampl (µV) | TB-Ampl (µV) | Click-Ampl (µV) |
| 37 | 58 | F | R | 5.4 | 20.00 | 0.28 | 0.97 | 0.83 | 75 | 251.0 | - | 95 | 13.0 | - | - |
| 38 | 58 | F | L | 4.1 | 23.33 | 0.43 | 0.81 | 0.76 | 70 | 321.0 | - | 85 | 16.6 | - | - |
| 39 | 72 | F | L | 4.4 | 18.33 | 0.56 | 0.90 | 0.83 | 85 | 257.0 | - | 100 | 14.2 | - | - |
| 40 | 56 | F | R | 4.3 | 30.00 | 0.59 | 1.21 | 0.81 | 65 | 348.0 | - | 75 | 62.8 | - | - |
| 41 | 65 | F | L | 3.0 | 11.67 | 0.31 | 0.81 | 0.78 | 90 | 224.0 | - | 90 | 16.0 | - | - |
| 42 | 74 | M | R | 5.3 | 28.33 | 0.55 | 0.89 | 0.78 | 75 | 130.0 | - | 85 | 48.0 | - | - |
| 43 | 72 | M | R | 3.9 | 16.67 | 0.54 | 0.88 | 1.27 | 100 | 93.0 | - | 100 | 22.7 | - | - |
| 44 | 92 | M | R | 2.1 * | 8.33 | 0.54 | 1.05 | 0.75 | 100 | 104.0 | - | 100 | 15.7 | - | - |
| 45 | 71 | F | L | 3.0 | 16.67 | 0.67 | 0.95 | 0.91 | 95 | 107.0 | - | 95 | 34.0 | - | - |
| 46 | 64 | M | R | 4.8 | 15.00 | 0.68 | 0.87 | 0.89 | 85 | 110.3 | - | 100 | 8.3 | - | - |
| Median | 65 | | | 4.1 | 21.67 | 0.52 | 1.00 | 0.80 | 85 | 177.0 | 200.8 | 95 | 24.4 | 13.3 | 13.2 |
| IQR | 15 | | | 1.5 | 13.33 | 0.17 | 0.25 | 0.14 | 15 | 151.8 | 130.5 | 15 | 21.7 | 12.5 | 12.7 |
| Range | 56 | | | 4.6 | 63.66 | 0.56 | 0.71 | 0.59 | 65 | 347.5 | 228.0 | 60 | 70.7 | 46.2 | 43.2 |

Abbreviations: ABG: air–bone gap. AC: air-conducted. Ampl: amplitude. BC: bone-conducted. cVEMPs: cervical vestibular-evoked myogenic potentials. F: female. HSC: horizontal semicircular canal. IQR: interquartile range. L: left. M: male. oVEMPs: ocular vestibular-evoked myogenic potentials. PSC: posterior semicircular canal. R: right. SSC: superior semicircular canal. SCD: superior canal dehiscence. TB: tone bursts. Thres: threshold. vHIT: video head impulse test. VOR: vestibulo-ocular reflex. (*) means that the dehiscence was due to a deep groove of the superior petrosal sinus.

**Table 2.** Demographics and pre-operative symptoms and signs of each patient with SCD who underwent effective SSC plugging for disabling symptoms and was included in the analysis (n = 10).

| Ear | Age | Sex | SCD Side | Pre-Operative Auditory Symptoms and Signs | | | | | Pre-Operative Vestibular Symptoms and Signs | | | |
|---|---|---|---|---|---|---|---|---|---|---|---|---|
| | | | | Hearing | Hyperacusis | Autophony | Pulsatile Tinnitus | Fullness | Vertigo/Unsteadiness | SVIN Test | vHIT Findings | c/o VEMPs |
| 1 | 19 | M | R | mixed HL | - | yes | yes | - | recurrent vertigo, T, H | up/contra ny | normal | enhanced |
| 2 | 49 | F | L | normal | - | yes | yes | yes | unsteadiness, T, H | up/contra ny | normal | enhanced |
| 3 | 65 | M | R | mixed HL | - | yes | yes | yes | unsteadiness, T, H | ipsi ny | SSC impairment | enhanced |
| 4 | 48 | F | L | CHL | yes | yes | yes | yes | recurrent vertigo | down/ipsi ny | SSC impairment | enhanced |
| 5 | 39 | M | R | CHL | yes | yes | yes | yes | recurrent vertigo, T, H | up/contra ny | SSC impairment | enhanced |
| 6 | 74 | F | Bil | CHL | - | - | - | L | recurrent L BPPV, T, H | L ny | SSC impairment | enhanced |
| 7 | 45 | F | Bil | normal | L | L | Bil | L | unsteadiness, recurrent L BPPV, T, H | down/L ny | normal | enhanced |
| 8 | 54 | M | L | mixed HL | yes | yes | yes | yes | T, H | up/contra ny | normal | enhanced |
| 9 | 68 | M | L | CHL | yes | yes | yes | yes | unsteadiness, T | contra ny | SSC impairment | enhanced |
| 10 | 40 | M | L | normal | yes | yes | yes | yes | H | up/contra ny | SSC impairment | enhanced |

Abbreviations: Bil: bilateral. BPPV: benign paroxysmal positional vertigo. c: cervical. CHL: conductive hearing loss. contra: contralesional. down: downbeating. F: female. H: Hennebert sign. HL: hearing loss. ipsi: ipsilesional. L: left. M: male. ny: nystagmus. o: ocular. R: right. SSC: superior semicircular canal. SCD: superior canal dehiscence. SVIN: skull vibration-induced nystagmus. T: Tullio phenomenon. up: upbeating. vHIT: video head impulse test. VEMPs: vestibular-evoked myogenic potentials. Where not specified, data reported refer to the ear that was plugged.

**Table 3.** Demographics and post-operative symptoms and signs recorded within the first 2 weeks from surgery of each patient with SCD who underwent SSC plugging for disabling symptoms and was included in the analysis (n = 10).

| Ear | Age | Sex | Side of Plug | Post-Operative Auditory Symptoms and Signs | | | | | Post-Operative Vestibular Symptoms and Signs | | | |
|---|---|---|---|---|---|---|---|---|---|---|---|---|
| | | | | Transient Hearing Loss | Hyperacusis | Autophony | Pulsatile Tinnitus | Fullness | Transient Vertigo/ Unsteadiness | Transient Spontaneous ny | Transient vHIT Findings | c/o VEMPs |
| 1 | 19 | M | R | slight flat SNHL | - | improved | improved | - | unsteadiness | contra | - | absent |
| 2 | 49 | F | L | slight flat SNHL | - | improved | improved | improved | unsteadiness + BPPV | ipsi then contra | global SC impairment | reduced |
| 3 | 65 | M | R | slight flat SNHL | - | improved | improved | improved | unsteadiness + BPPV | ipsi/down then contra | global SC impairment | absent |
| 4 | 48 | F | L | - | improved | improved | improved | improved | unsteadiness | ipsi then contra | global SC impairment | absent |
| 5 | 39 | M | R | - | improved | improved | improved | improved | unsteadiness | ipsi | - | reduced |
| 6 | 74 | F | L | severe flat SNHL | - | - | - | improved | unsteadiness | ipsi then contra | global SC impairment | reduced |
| 7 | 45 | F | L | - | improved | improved | improved | improved | unsteadiness + BPPV | down/ipsi then contra | global SC impairment | reduced |
| 8 | 54 | M | L | - | improved | improved | improved | improved | unsteadiness + BPPV | up/contra ny | - | reduced |
| 9 | 68 | M | L | severe flat SNHL | improved | improved | improved | improved | unsteadiness | ipsi then contra | global SC impairment | absent |
| 10 | 40 | M | L | - | improved | improved | improved | improved | unsteadiness + BPPV | ipsi | - | reduced |

Abbreviations: BPPV: benign paroxysmal positional vertigo. c: cervical. contra: contralesional. down: downbeating. F: female. ipsi: ipsilesional. L: left. M: male. ny: nystagmus. o: ocular. R: right. SC: semicircular canal. SCD: superior canal dehiscence. SNHL: sensorineural hearing loss. up: upbeating. vHIT: video head impulse test. VEMPs: vestibular-evoked myogenic potentials. Where not specified, data reported refer to the ear that was plugged.

**Table 4.** Demographics and instrumental findings at 6 months for each ear in the "post-operative SCD" group included in the analysis (n = 10).

| Ear | Age | Sex | Side of SCD Surgery | SCD Size (mm) | ABG (dB) | VOR-Gain on vHIT | | | AC cVEMPs | | BC cVEMPs | AC oVEMPs | | BC oVEMPs | |
|---|---|---|---|---|---|---|---|---|---|---|---|---|---|---|---|
| | | | | | | SSC | HSC | PSC | Thres (dB SPL) | Ampl (μV) | Ampl (μV) | Thres (dB SPL) | Ampl (μV) | TB-Ampl (μV) | Click-Ampl (μV) |
| 1 | 19 | M | R | 1.8 | 5.00 | 0.44 | 1.00 | 0.71 | 130 | 0.5 | - | - | - | - | - |
| 2 | 49 | F | L | 3.0 | 6.67 | 0.36 | 0.96 | 0.58 | 110 | 49.5 | - | - | - | 2.8 | 3.0 |
| 3 | 65 | M | R | 3.1 | 20.00 | 0.58 | 1.11 | 0.42 | 130 | 0.5 | 0.5 | 130 | 0.2 | 0.4 | 0.4 |
| 4 | 48 | F | L | 2.5 * | 11.67 | 0.66 | 0.86 | 0.85 | 120 | 29.0 | 43.5 | 125 | 3.2 | 3.8 | 0.8 |
| 5 | 39 | M | R | 4.0 | 21.67 | 0.47 | 1.01 | 0.80 | 120 | 71.0 | 106.5 | 130 | 0.2 | 0.8 | 5.2 |
| 6 | 74 | F | L | 4.5 | 21.67 | 0.30 | 0.98 | 0.51 | 120 | 15.5 | 61.0 | 130 | 0.2 | 1.4 | 10.8 |
| 7 | 45 | F | L | 1.8 | 10.00 | 0.23 | 1.15 | 0.61 | 110 | 61.5 | 55.5 | 120 | 5.2 | 5.0 | 4.8 |
| 8 | 54 | M | L | 3.0 | 23.33 | 0.65 | 0.99 | 0.69 | 95 | 112.0 | 102.0 | 100 | 22.0 | 16.8 | 12.2 |
| 9 | 68 | M | L | 5.0 | 16.67 | 0.47 | 0.85 | 0.50 | 120 | 47.0 | 168.0 | 130 | 0.2 | 2.2 | 5.4 |
| 10 | 40 | M | L | 4.3 | 6.67 | 0.37 | 0.98 | 0.82 | 115 | 39.5 | 115.5 | 120 | 2.6 | 1.4 | 0.6 |
| Median | 49 | | | 3.1 | 14.17 | 0.46 | 0.96 | 0.65 | 120 | 43.3 | 81.5 | 128 | 1.4 | 2.2 | 4.8 |
| IQR | 26 | | | 2.0 | 15.00 | 0.25 | 0.10 | 0.30 | 13 | 52.1 | 66.8 | 10 | 4.5 | 3.3 | 7.4 |
| Range | 55 | | | 3.2 | 18.33 | 0.43 | 0.30 | 0.43 | 35 | 111.5 | 167.5 | 30 | 21.8 | 16.4 | 11.8 |

Abbreviations: ABG: air–bone gap. AC: air-conducted. Ampl: amplitude. BC: bone-conducted. cVEMPs: cervical vestibular-evoked myogenic potentials. F: female. HSC: horizontal semicircular canal. IQR: interquartile range. L: left. M: male. oVEMPs: ocular vestibular-evoked myogenic potentials. PSC: posterior semicircular canal. R: right. SSC: superior semicircular canal. SCD: superior canal dehiscence. TB: tone bursts. Thres: threshold. vHIT: video head impulse test. VOR: vestibulo-ocular reflex. (*) means that the dehiscence was due to a deep groove of the superior petrosal sinus. Where not specified, data reported refer to the ear that was plugged.

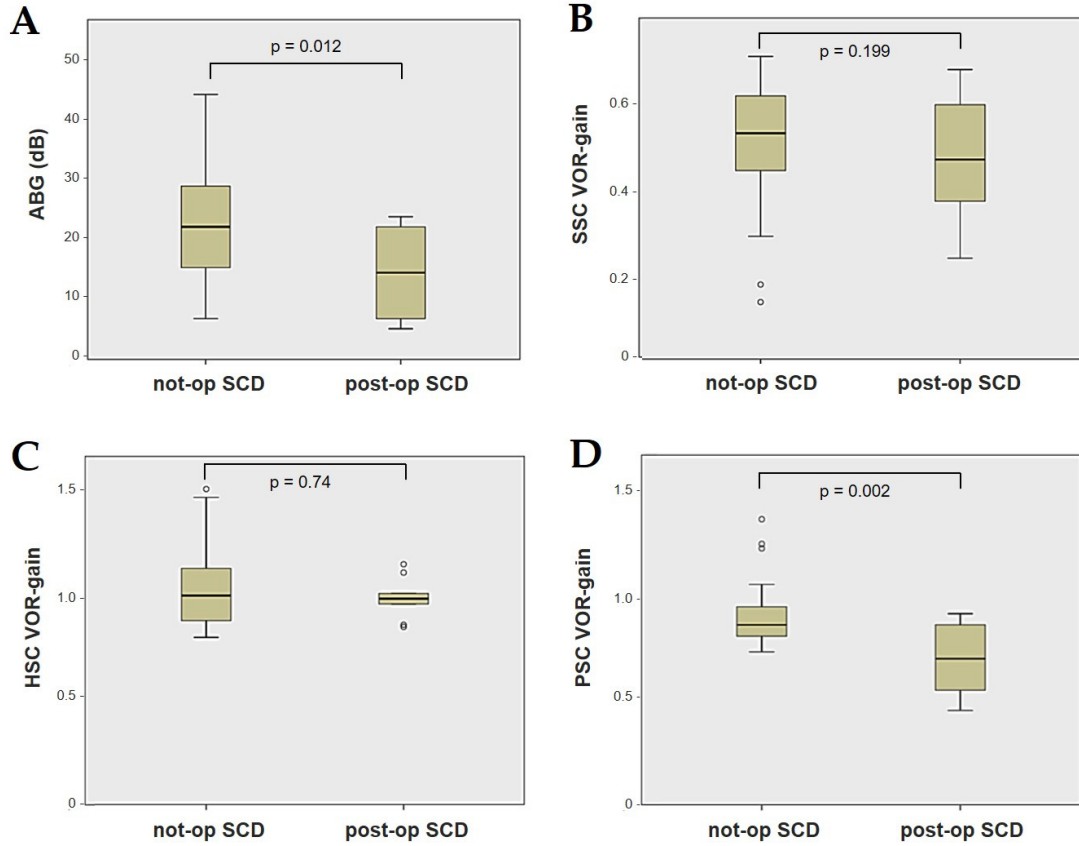

**Figure 2.** Box plots correlating ABG (**A**), SSC VOR gain (**B**), HSC VOR gain, (**C**) and PSC VOR gain (**D**) between not-operated ears with SCD and ears with SCD after surgical plug, with corresponding *p* value. ABG: air–bone gap. HSC: horizontal semicircular canal. Not-op: not-operated ears. Post-op: post-operative ears. PSC: posterior semicircular canal. SCD: superior canal dehiscence. SSC: superior semicircular canal.

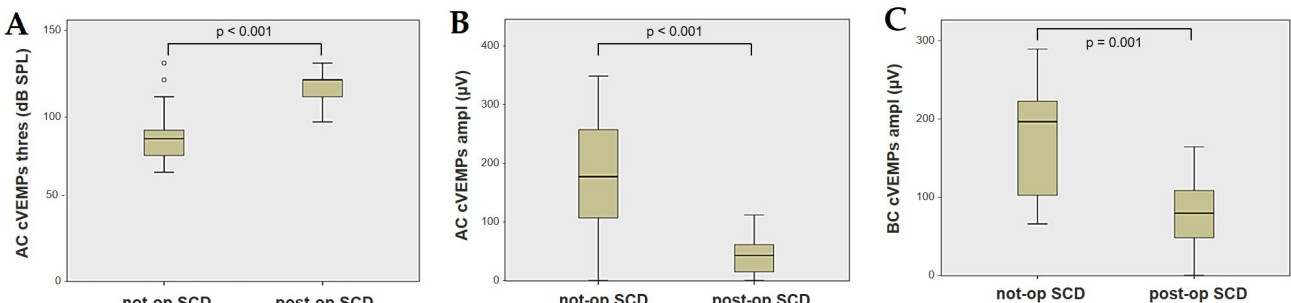

**Figure 3.** Box plots correlating AC cVEMPs threshold (**A**), AC cVEMPs amplitude, (**B**) and BC cVEMPs amplitude (**C**) between not-operated ears with SCD and ears with SCD after surgical plug, with corresponding *p* value. AC: air-conducted. Ampl: amplitude. BC: bone-conducted. cVEMPs: cervical vestibular-evoked myogenic potentials. Not-op: not-operated ears. Post-op: post-operative ears. SCD: superior canal dehiscence. Thres: threshold.

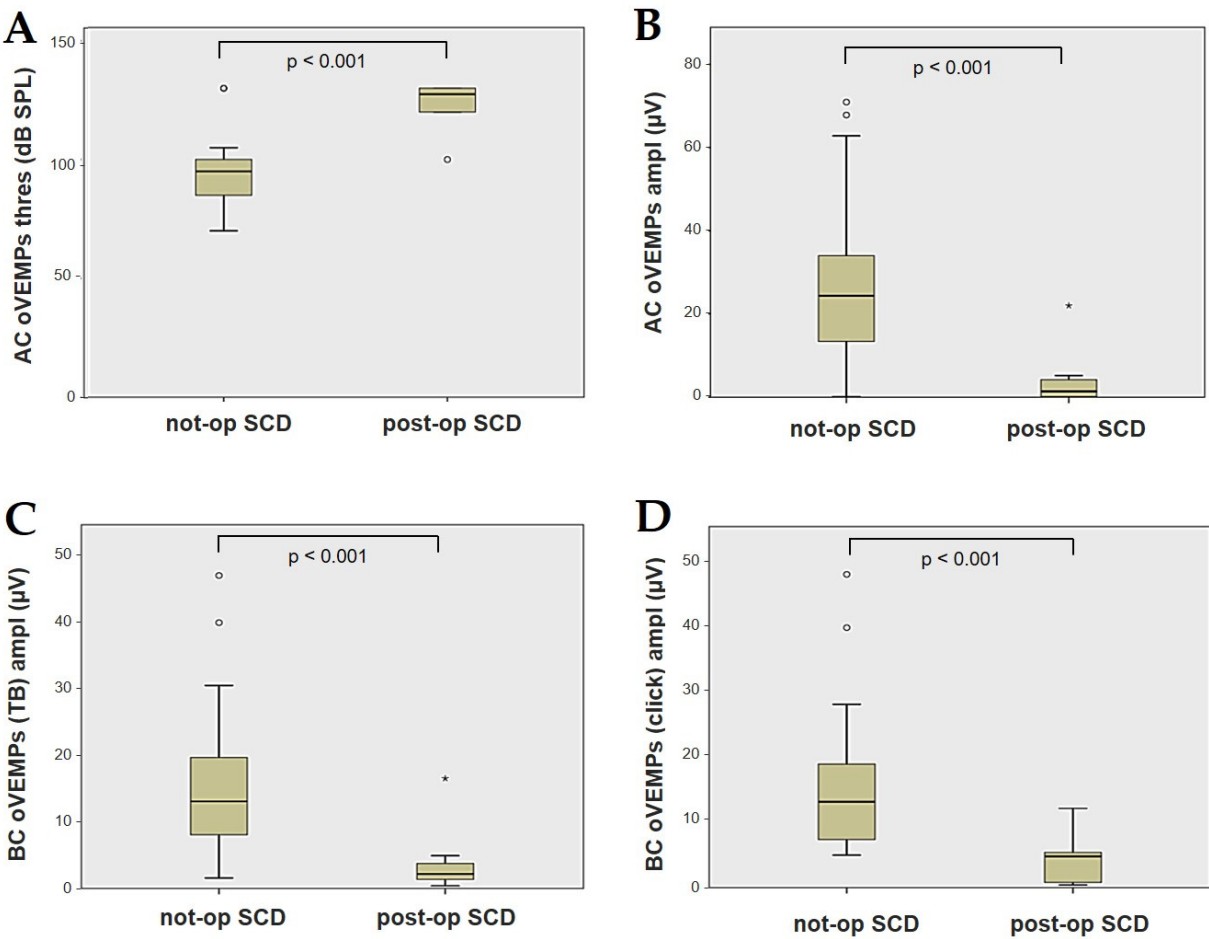

**Figure 4.** Box plots correlating AC oVEMPs threshold (**A**), AC oVEMPs amplitude, (**B**) and BC oVEMPs amplitude for tone burst (**C**) and for click stimuli (**D**) between not-operated ears with SCD and ears with SCD after surgical plug, with corresponding *p* value. AC: air-conducted. Ampl: amplitude. BC: bone-conducted. Not-op: not-operated ears. oVEMPs: ocular vestibular-evoked myogenic potentials. Post-op: post-operative ears. SCD: superior canal dehiscence. TB: tone bursts. Thres: threshold.

## 4. Discussion

According to our data, the ears that received a successful surgical SSC plug with improvement of disabling symptoms at 6 months developed an extremely different instrumental profile compared to the ears with SCD presenting with reduced VOR gain values for the affected SSC. The two cohorts only shared the same VOR gain reduction in the SCC in the vHIT. In particular, patients with SCD and hypoactive SSC mostly presented with an array of signs consistent with a TMWM, as if the dehiscence was particularly patent to allow a dissipation of a significant amount of endolymphatic flow. In fact, low-frequency CHL, enhanced amplitudes, and reduced thresholds of VEMPs to AC and BC stimuli have been demonstrated to occur as a result of reduced intralabyrinthine impedance due to the bony defect [3–7]. All of them presented with different degrees of selective hypofunction of the affected SSC in the vHIT, while none of them developed HSC or PSC functional loss, confirming the close relationship between SCD and a VOR gain deficiency, as already demonstrated in the literature [2,9,26,29]. In these patients, we only focused on the instrumental profile of ears with SCD without considering the symptoms, as it would have gone beyond the aims of our study. On the contrary, all patients included in the "post-operative SCD" group developed a reduction in symptoms after surgery, and SSC plugging resulted in the normalization and/or reduction of pre-operative abnormal instrumental findings.

Namely, CHL was restored (<15 dB) in half of the cases, while a residual ABG ≥15 dB could still be detected in half of them. An→ SNHL, either transient or persistent, was detected in a significant number of patients, consistent with data available in the literature [3,11–14,16–18,22,25,54,55]. Overall, the VEMP threshold normalized and amplitudes decreased, which was consistent with a restoration of the physiological impedance of the inner ear [10,13,21,23,25]. A significant subgroup of patients who underwent surgery developed a global vestibular hypofunction likely due either to perilymphatic leakage or a distension of the contiguous partitions of the membranous labyrinthine on a transient basis, as was already reported in the post-operative days for patients with a wide-sized SCD [9,13,25,56–58]. Interestingly, all patients showed either ipsilesional or contralesional spontaneous nystagmus with variable torsional components in the first days after surgery, which exhibited a spontaneous reversal of direction in the following days in six of the patients, behaving similarly to MD [56]. Even the onset of both SNHL and ipsilesional PSC-BPPV in the post-operative period supports the hypothesis of a transitory perturbation of the whole inner ear, leading to damage to the cochlear hair cells and dislodgement of otoconia from the utricle, respectively [22,54,59]. While HSC-VOR gain fully recovered in all cases, PSC function restored only in some of them. Even though it has been reported in the literature how the SSC function itself might recover over time [21,29], all our patients still exhibited a selective VOR reduction for the SSC at 6 months, consistent with the long-term effect of surgical plugging [9,25,56,58,60]. This discrepancy in recovery patterns of the SSC-VOR gain among studies may be attributable to the different degrees of functional deterioration immediately after surgery, likely due to the different materials and approach (transmastoid versus middle fossa) that were adopted, the variable sample sizes that were analyzed, and the heterogeneous follow-up period. On the other hand, studies on animal models have demonstrated how canal plugging is effective in attenuating sensitivity to head rotation only at low frequencies while canal function can be preserved at high-acceleration stimuli, likely due to the onset of cupular displacements generated by local deformation of the membranous canal [61,62]. Nevertheless, overall post-operative measures are in line with the literature concerning audio-vestibular findings after a surgical canal plug in SCD, namely, after restoring a physiological two-window system.

In light of the aforementioned findings, it is less likely that the SSC VOR gain reduction in patients with SCD presenting with a clinical/instrumental profile consistent with TMWM could be ascribed to a natural "spontaneous plugging" process, as has been speculated so far [2,9,27]. Therefore, we aimed to identify an alternative mechanism accounting for the vHIT findings in SCD. An endolymphatic flow dissipation at the dehiscence during head impulses has been recently proposed as an alternative putative mechanism [52]. In other words, while in normal conditions the bony canal integrity allows the high-acceleration flow energy to be correctly driven along the plane of the tested canal through head impulses, a canal dehiscence could likely dissipate part of this energy. The lack of the bony roof of the SCC would result in an altered balance of the fluid dynamics during high-acceleration head impulses along the SCC plane, resulting in a transient membranous canal indentation. The same process has been demonstrated to activate both regular and irregular afferents of the SSC in response to loud sounds, pressure stimuli, and skull vibrations [5–8]. This condition might lead to an endolymphatic flow dissipation, which could result in a reduced amount of fluid-mechanical wave efficiently pulling the SSC cupula away from the utricle to excite canal afferents during downward head impulses, thus accounting for an "apparent" VOR gain hypofunction (or VOR gain "pseudo-hypofunction") on vHIT. Another finding that strongly supports the assumption that the SSC is patent and dehiscent in cases of SCD with SSC VOR gain reduction is the enhanced amplitude of both cVEMPs and oVEMPs to AC and BC sounds. In fact, according to studies on animal models, the augmented VEMP responses in SCD patients are probably due to the compound action of the irregular afferents of the SSC and otolith organs, which have been demonstrated to be activated only in cases of SCD, while being silent in cases of normally encased SSC [6,7]. Therefore, the same low-impedance pathway generating symptoms and signs consistent with a TMWM, such as

low frequency ABG, nystagmus evoked by sound/pressure stimuli or skull vibrations, and enhanced VEMPs with AC/BC sounds, could also account for the VOR gain hypofunction involving the dehiscent canal. This hypothesis can also explain why large-sized SCDs (where a stronger TMWM is expected to occur) are more prone to generate sound/pressure nystagmus, abnormal VEMPs, a wider ABG, and lower SSC VOR gain values compared to small-sized defects [2,6,9,10,26,29,40–51]. Additional factors might account for different results: the SCD location; the membranous canal impedance; the different degrees of compliance of the surrounding structures other than the dura which can be in contact with the membranous SSC, such as the SPS; and even additional transient/persistent factors, such as idiopathic intracranial hypertension, "Apparent" canal VOR gain reduction could therefore represent an additional typical sign of TMWM in case of a third window lesion involving a semicircular canal. Nevertheless, despite SSC hypofunction representing a common finding in SCD, the Barany Society diagnostic criteria does not include vHIT among the instrumental battery [53]. In our opinion, the guidelines for the diagnosis of SCD should consider vHIT data as a key finding for the diagnosis of third window disorders. Future studies involving a greater number of cases or research on animal models are needed to strengthen the validity of our considerations. Accordingly, it should also be remarked that the reduction of SSC VOR gain at the vHIT should not be taken as a certain marker of canal blockage after canal plugging in SCD, for all the aforementioned considerations. Therefore, the diagnostic value of canal VOR gain reduction on vHIT as a marker of restoration of a physiologic two-window system in the inner ear appears to be weaker than thought before.

Even though our data seem to support an endolymphatic flow dissipation during head impulses as the putative pathomechanism behind the SSC VOR decrease in SCD rather than a "spontaneous plugging" process, some considerations still need to be made. First, it is reasonable to assume that the middle fossa dura might not be as effective in plugging the SSC as the materials that are commonly used in surgery. In fact, the natural dural plug might be incomplete or intermittent, such that endolymphatic flows that are produced by sound and pressure stimuli could be strong enough to overcome the effect of plugging and generate an abnormal stimulation of the canal afferents [9,27]. Nevertheless, if that were the case, patients should be expected to develop partly mitigated symptoms, or atypical symptoms for a TMWM, and/or near-normal instrumental findings, but it was not the case for the majority of the patients included in this study. On the other hand, according to the inclusion criteria adopted in our study, we excluded from the correlation analysis the patients with SCD presenting with a clinical scenario consistent with MD, aiming to avoid possible interference in data collection due to additional pathologies other than SCD itself. Nevertheless, the literature reporting how third window disorders might result in MD is increasing, broadening the range of the clinical phenotypes related to TMWM [63–67]. In fact, it could be speculated that an intermittent spontaneous prolapse of the dura within the canal might occur in this subgroup of patients, resulting in a transient displacement of the contained volume of the endolymph from the SSC into the contiguous areas, thus developing a transitory distension of the residual membranous labyrinth in a manner comparable to endolymphatic hydrops. Similarly, a transitory hydropic state of the inner ear has already been proposed to explain a transient spontaneous irritative nystagmus and hearing impairment detectable in the first post-operative hours [56,68], as was the case for a subgroup of patients in our series. Therefore, we might have excluded from the analysis those few patients who likely developed a "spontaneous plugging" process. Similar speculative considerations could be proposed in cases with SCD without symptoms: here, a progressive dural prolapse might have slowly occurred allowing compensation mechanisms to prevent the onset of symptoms. Finally, as for the "not-operated SCD" group, it should be underlined that not all ears exhibited a complete audio-vestibular profile fully consistent with TMWM. In fact, 10 ears presented with an almost negligible ABG (<15 dB) on audiometry and 12 ears exhibited near-normal c/oVEMP findings (threshold $\geq$100 dB SPL and/or either near-normal or reduced amplitudes). Even though, as already

mentioned, it is well known how several factors might account for the variability of signs, including SCD size and location, it could also be hypothesized that in some of these patients a partial "spontaneous plugging" process might actually occur, resulting in an incomplete instrumental profile. Nevertheless, it was not possible to visualize a potential spontaneous natural plugging of the affected SSC in these patients, as 3D-inner ear MRI sequences were not systematically included in our working methodology. Moreover, the evaluation of same sequences would have been beneficial in the interpretation of symptoms of patients who developed a significant SNHL after surgical plugging, where a possible post-operative hydrops of the contiguous labyrinthine structures might have occurred [56,68,69]. Furthermore, we only considered the instrumental data of the ears with SCD without focusing on the symptoms of patients. It might be speculated that patients presenting with an incomplete instrumental profile tend to develop only mitigated symptoms, consistent with a partial natural plug of the dura resulting in a mitigated TMWM. Nevertheless, data on the possible natural evolutions of SCD are still lacking in the literature, as are data on asymptomatic SCD patients [70–73]. Future research and targeted studies are needed to confirm these correlations, with the aim to propose an improved method for targeted treatments based on different SCD profiles.

## 5. Conclusions

According to our data, VOR gain reduction for the affected canal in SCD is hardly explainable through a "spontaneous plugging" process in patients presenting with symptoms and signs consistent with TMWM. A dissipation of endolymphatic flows at the dehiscence that is likely due to an indentation of the membranous canal during head impulses can be proposed as an alternative pathomechanism accounting for an "apparent" SSC hypofunction. VOR gain impairment for the dehiscent canal should be considered as an additional sign of a TMWM.

**Author Contributions:** Conceptualization, A.C., M.A., E.A., P.M. and S.M.; methodology, A.C.; validation, L.P. and A.G.; formal analysis, A.C.; investigation and data collection, A.C., C.B., G.P., R.R.R. and P.B.; writing—original draft preparation, A.C.; writing—review and editing, M.A., E.A., P.M. and S.M.; supervision, L.P. and A.G. All authors have read and agreed to the published version of the manuscript.

**Funding:** This research received no external funding.

**Institutional Review Board Statement:** The study was conducted in accordance with the Declaration of Helsinki, and approved by the Institutional Review Board of Area Vasta Emilia Centro (protocol code 143/2019/Oss/AOUBo, date of approval 8 April 2019).

**Informed Consent Statement:** Informed consent was obtained from all subjects involved in the study.

**Data Availability Statement:** The data presented in this study are available on request from the corresponding author. The data are not publicly available due to ethical restrictions.

**Acknowledgments:** In memory of Giovanni Carlo Modugno and Erik Ulmer.

**Conflicts of Interest:** The authors declare no conflict of interest.

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
