# Peer review of "Impaired Vestibulo-Ocular Reflex on Video Head Impulse Test in Superior Canal Dehiscence: “Spontaneous Plugging” or Endolymphatic Flow Dissipation?"

_audiolres, doi:10.3390/audiolres13050071_

Round 1

Reviewer 1 Report

This is a well-written manuscript, relevant at least for opening a discussion on the natural evolution of CSS dehiscence. This is because, so far, there are only a few references in the literature about the natural (spontaneous) self-plugging process of this variant of Third Mobile Window Abnormality and its intimate anatomopathological mechanism is still unknown. I think that here the authors could hypothesize that once we better understand this phenomenon, we may be able to propose an improved method of surgical treatment, given that in many cases of SCD with autoplugging these patients are much less symptomatic - a fact also mentioned by the authors.

The merit of this manuscript also lies in the fact that it discusses something omitted from the consensus criteria for a positive diagnosis of symptomatic SSCD by the Barany Society. Namely that RVO is often reduced by the dehiscent CSS side, and this is a point that should be clarified in future studies. The authors also propose a pathophysiological hypothesis for this diminished gain of RVO on the high stimulation frequencies of the CSS involved in a DCS tested by vHIT , which I personally consider plausible, still difficult to verify.

Although the audiological and vestibular examinations have been well, even exhaustively presented, I consider as a criticizable lack of this article the fact that the authors did not include in their working methodology the performance of a 3D labyrinthine MRI to visualize - at least in some cases - the possible spontaneous self-plugging of DCS especially in those patients in whom the authors had the clinical “feeling” that it may exist. The same for those cases of significant sensorineural deafness post CSS plugging surgery, where there were no fears of a surgical technique default. This is more so as there is a very recent reference on this pathology treating about the eventual sources of surgical failure including post plugging hydrops SSCD which the authors also consider in this paper. (https://doi.org/10.3389/fneur.2023.1209567 ). 

Despite this important but singular criticism, I believe the manuscript has the necessary qualities to be published. Perhaps it would be better if the authors specified in the "limits of the paper" this fact.

Minor comment: the authors should rather use throughout the manuscript the abbreviation "TMWM" - third mobile window mechanism - instead of "TWM" 

No comments.

Author Response

We thank very much the Editor of “Audiology Research” and all Reviewers for having considered our manuscript. We sincerely appreciate Reviewers’ valuable and detailed comments and strongly believe the manuscript has been substantially improved by these suggestions.

A response to all your valuable comments is reported below (in bold), whereas the changes are highlighted in yellow in the revised copy of the manuscript.

We hope to have correctly corrected all unclear and wrong aspects throughout the text and hope that our revised manuscript may have addressed all comments.

Reviewer 2 Report

This is a retrospective review of symptomatic SCDS cases with impaired SSC VOR gain compared with SCDS cases surgically plugged. The aim of the study is to highlight the presence of impaired SSC gain in not operated SCDS ears and to interpret this finding in respect to the other instrumental findings. The study discusses an alternative interpretation of this SCDS instrumental profile, with respect to the since before accredited canal natural plugging: the canal flow dissipation.  

Overall, the study is well done, interesting and relevant for the research field. It is for my taste a little bit too speculative in the discussion section, as I have noted in the attached file. Otherwise It sounds original and relevant. 

Please refer to the attached file with comments. 

Good english writing.

Author Response

(The authors gave the same response as above.)

Reviewer 3 Report

Surgical plugging of the SCC has been present for a while but is not vastly adopted due to the controversial surgical outcomes and morbidity. The set up of the study and the presentation of the results are sufficient. However, this study focuses to measurements, not mentioning symptoms of the patients pre or post-op lacking clinical correlation of the results. 

Some minor English language errors - will require another language review to ensure these are corrected

Author Response

(The authors gave the same response as above.)
